# Machine learning for cell type classification from single nucleus RNA sequencing data

Huy Le[1], Beverly Peng[1], Janelle Uy[1], Daniel Carrillo[1], Yun Zhang[2], Brian D. Aevermann[2], Richard H. Scheuermann [2,3,4]*

1 Department of Bioengineering, University of California, San Diego, CA, United States of America, 2 Department of informatics, J. Craig Venter Institute, La Jolla, CA, United States of America, 3 Department of Pathology, University of California, San Diego, CA, United States of America, 4 La Jolla Institute for Immunology, San Diego, CA, United States of America

* rscheuermann@jcvi.org

**Data Availability Statement:** The dataset can be found at the Allen Brain Map at https://portal.brain-map.org/atlases-and-data/rnaseq/human-mtg-smart-seq. The code used to preprocess the data and run each model can be found in this GitHub

## Abstract

With the advent of single cell/nucleus RNA sequencing (sc/snRNA-seq), the field of cell phenotyping is now a data-driven exercise providing statistical evidence to support cell type/state categorization. However, the task of classifying cells into specific, well-defined categories with the empirical data provided by sc/snRNA-seq remains nontrivial due to the difficulty in determining specific differences between related cell types with close transcriptional similarities, resulting in challenges with matching cell types identified in separate experiments. To investigate possible approaches to overcome these obstacles, we explored the use of supervised machine learning methods—logistic regression, support vector machines, random forests, neural networks, and light gradient boosting machine (LightGBM)–as approaches to classify cell types using snRNA-seq datasets from human brain middle temporal gyrus (MTG) and human kidney. Classification accuracy was evaluated using an F-beta score weighted in favor of precision to account for technical artifacts of gene expression dropout. We examined the impact of hyperparameter optimization and feature selection methods on F-beta score performance. We found that the best performing model for granular cell type classification in both datasets is a multinomial logistic regression classifier and that an effective feature selection step was the most influential factor in optimizing the performance of the machine learning pipelines.

## Introduction

Next-generation sequencing (NGS) with high throughput technologies has radically transformed biomedical research, providing novel avenues to analyze the foundations of biological systems. These technologies have led to an exponential growth in high-content biological datasets, challenging scientists to come up with robust computational tools to aid in their processing and analysis [1]. One newly emerging application of NGS is to measure the gene expression profiles in individual cells using single cell/nucleus RNA sequencing (sc/snRNA-seq) in order to characterize the transcriptional phenotypes of individual cells in complex

repository: https://github.com/HuyGLe/neuronal-cell-type-classification.

**Funding:** We would also note that the work reported in this manuscript was funded by the U.S. National Institutes of Health (RF1MH123220). The funding bodies had no role in the design or conclusions of this study.

**Competing interests:** The authors have declared that no competing interests exist.

tissues [2]. sc/snRNA-seq has been used to investigate a number of tissues including ileum, colon, rectum, kidney, liver, pancreas, heart, lung, prostate, testis, placenta, skin, eye, and blood [3]. For example, snRNA-seq allowed for the identification of a specialized "rosehip" GABAergic neuron in human cortical layer 1 of the middle temporal gyrus with gap junctions that target apical dendritic shafts of layer 3 pyramidal neurons to inhibit back propagation of action potentials [4]. A more comprehensive snRNA-seq study of the entire thickness of the MTG neocortex identified 75 transcriptomically distinct cell types, including 45 GABAergic interneurons and 24 excitatory glutamatergic neurons, with cortical layer-specific distributions [5]. These examples illustrate the power of single cell transcriptional profiling to effectively characterize the cellular diversity of human neuronal tissues.

A typical sc/snRNA-seq data analysis workflow includes steps for reducing the number of gene features, creating a manifold representation for visualization, unsupervised clustering to define discrete cell types, and differential expression analysis to identify expression signatures of cell populations. Common tools used to do so include Seurat, Scanpy, Pagoda, and scVi [6–9]. These methods have different ways of selecting highly variable genes, which are then used in subsequent cell type/state characterization and/or trajectory analysis [10].

While these methods have been crucial in identifying novel cell types and uncovering unknown cell-gene relationships, manual annotation of these cell clusters is both time-consuming and subjective. In addition, it can often be challenging to generalize and project these cell type annotations to a new experiment. Several statistical methods have recently been reported that attempt to match cell type clusters between experiments by quantifying the similarity of the gene expression profiles of cells in the clusters being compared [11]. For example, the FR-Match algorithm uses a graph-theoretic method based on the non-parametric multivariate Friedman-Rafsky test to quantify gene expression similarities [12]. Seurat's Azimuth platform utilizes reference-based canonical correlation analysis to map scRNA-seq datasets [13].

In this study, we have re-envisioned cell type matching as a classification problem in order to explore the use of machine learning for scRNA-seq cell type cluster matching. Machine learning, a subset of artificial intelligence in which computers are trained to improve on tasks with experience, is an example of a computational tool with the ability to discover patterns in large datasets and construct predictive classification models based on these patterns. Machine learning applications have become popular in the biomedical research community and are now being incorporated into many genomics, proteomics, metagenomics, and systems biology workflows [14], including for cell type annotation from scRNA-seq experiments [15]. A recent benchmark study examined 13 popular supervised learning techniques for cell phenotype classification on 27 datasets of various sample sizes (from 91 cells to 37,464 cells) [16]. Their study suggested that ElasticNet with interactions was the best performer for small- and medium-sized datasets, with linear discriminant analysis (LDA) being the fastest algorithm for all datasets. Similarly, Abdelaal et al. benchmarked the performance of 22 classification methods, using the default settings established by the methods developers, on 27 mostly low-complexity (small numbers of cell populations) publicly-available datasets [17]. Many of the methods demonstrated excellent performance on these low-complexity datasets, with SVM with the rejection option ($SVM_{rejection}$) performing the best overall. In addition to the generic supervised machine learning models, there are also computational methods developed specifically for cell phenotype classification using single cell data. For example, scAnnotate is a supervised machine learning model that uses the proportion of dropouts as a useful feature selection metric to help classification [18]. However, few studies have focused on evaluating the performance of multiclass models used to classify cell types with high granularity, e.g., the 75 cell types in the human MTG dataset. The heterogeneities and similarities among the granular cell types present bigger challenges for the classification models. The primary focus of the study

reported here was to explore the effects of seven of the most popular binary and multiclass supervised classification methods (binary and multinomial logistic regression models, support vector machine (SVM), binary and multiclass random forest, gradient boosting machine (GBM), and neural networks), feature selection, and hyperparameter optimization on the performance of granular cell type classification.

## Materials and methods

Fig 1 provides an overview of the analysis workflow, including steps for dataset pre-processing, feature selection, machine learning model construction, hyperparameter optimization, and model prediction testing. Details are described in the subsequent sections.

### Brain dataset

The human middle temporal gyrus (MTG) snRNA-seq dataset [5] was retrieved from the Allen Brain Map (https://portal.brain-map.org/atlases-and-data/rnaseq) portal. For these data, sample nuclei were isolated from human MTG specimens utilizing Dounce homogenization and fluorescence-activated nuclei sorting, followed by the Smart-seq2 protocol to generate full length cDNA libraries for sequencing [19]. After data processing, the dataset consisted of 50,281 gene-level (exonic and intronic) read count values (expression estimates) for 15,928 samples taken from 8 human tissue donors between the ages of 24–66. Each nucleus was classified into one of 75 transcriptionally distinct cell types, including 45 inhibitory neuron types, 24 excitatory neuron types, and 6 non-neuronal types, using unsupervised iterative Louvain clustering in principal component analysis (PCA) space as described [20].

For the original study [5], the Western Institutional Review Board (WIRB) reviewed the use of de-identified postmortem brain tissue for research purposes and determined that, in accordance with federal regulation 45 CFR 46 and associated guidance, the use of and generation of

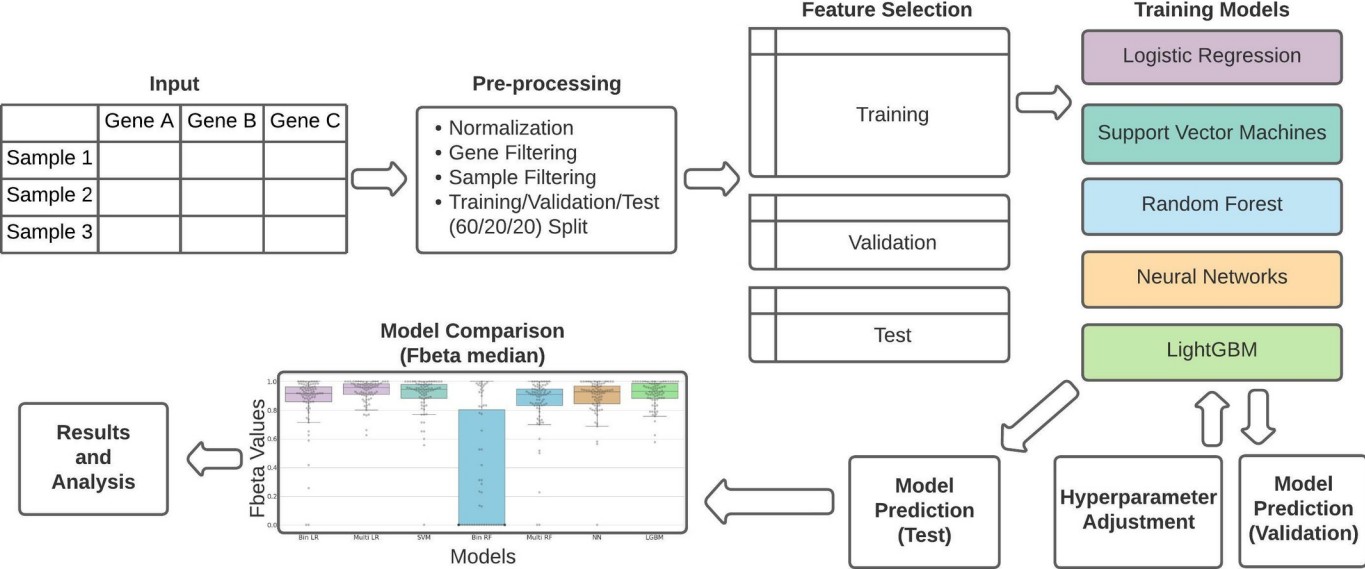

**Fig 1. Overview of the machine learning pipeline.** A count matrix undergoes pre-processing, including normalization and filtering. The data is randomly split into training (60%), validation (20%), and test (20%) sets independently for each cell type. The training sets are used to train the models. The validation set provides an initial test for accuracy of the trained models and is used to adjust the model's hyperparameters. Once the hyperparameters are optimized, the test set is run through each model and the F-beta score distribution across all clusters is used for model comparison.

data from de-identified specimens from deceased individuals did not constitute human subjects research requiring institutional review board review.

## Data pre-processing

The full snRNA-seq dataset was normalized to counts per million (cpm) and transformed using log2(cpm+1). The resulting matrix (50,281 genes x 15,928 nuclei) was filtered to remove nuclei without a cell-type assignment (50,281 genes x 15,603 nuclei) and to remove genes with zero expression across all samples (48,440 genes x 15,603 nuclei). The Median Gene Expression within each Cell Type (MGECT) was calculated for every gene. To remove possible housekeeping genes, genes with zero variance for their MGECT across all cell types were excluded, resulting in 13,945 genes x 15,603 samples. These housekeeping genes were expected to be of limited utility for developing predictive models that could distinguish between cell types because they had similar levels of expression across the different cell types. Nuclei from this filtered expression matrix were divided into training (60%), validation (20%), and test (20%) sets by stratified random sampling within each cluster. For all machine learning methods discussed in this paper, the training dataset was used to build the model, the validation dataset was used to select the best performing hyperparameters, and the test dataset was used to report the final model performance.

## Feature selection

Two separate feature selection metrics to select genes for model construction were explored: a binary expression score (BIN) and coefficient of variation (CV). The binary score is a metric developed to enrich for genes with an "all-or-none" expression pattern for each target cluster (cell type) with respect to the rest of the other clusters [21]. The binary score for gene g in cluster X is defined by the following equation:

$$Score_{g,X} = \frac{\sum_{i=1}^{n} (1 - \frac{y_i}{y_X})^+}{n - 1},$$

where $y_i$ is the median gene expression for cluster i, $y_X$ is the median expression in the target cluster X, n is the number of clusters, and $(\cdot)^+$ denotes the non-negative value of a real number. This results in a binary expression score in the range of [0, 1] with a score of 1 being the ideal case where the gene is only expressed in the target cluster. All genes were ranked according to their binary score for each cluster (see S1 Fig in S1 File for examples) and the top 1%, 5%, 10%, and 15% of genes from each cluster were merged into four final binary score feature sets. These percentage thresholds were chosen after manual inspection of the dataset sizes to allow for some feature reduction while not selecting too few genes from each cluster.

The second metric, coefficient of variation (CV), calculates the cross-cell-type variability of gene expression standardized by the mean expression and was performed on the MGECT found during the pre-processing step. We calculated CV using

$$CV = \frac{\sigma(MGECT)}{\mu(MGECT)},$$

where $\sigma(\cdot)$ and $\mu(\cdot)$ are the standard deviation and mean of a variable, respectively. Thresholds of 0.52, 1.5, 2.5, 3.5, and 4.5 were utilized to produce a broad range of overlapping gene subsets. When the distribution of CV of each gene is plotted, one peak can be seen around 0.52 (see S2 Fig in S1 File). The threshold minimum value was set to this peak value to exclude low CV values that were expected to provide limited classification utility. Subsequent threshold

values were chosen to evaluate F-beta performances as the threshold increased. The highest threshold value chosen was 4.5 as the F-beta performance for all models began to decrease dramatically above this value.

## Machine learning models

Using the previously defined 75 cell type classifications as labels, we input the filtered dataset into each of the supervised machine learning models: logistic regression, support vector machine (SVM), random forests, neural network, and LightGBM. Logistic regression, SVM, and random forests were built using scikit-learn packages [22]; neural networks were built using Tensorflow 2.0 [23]; LightGBM models were built using the LightGBM package [24].

Logistic regression is a model which applies the sigmoid function to a linear equation to output a range of values between 0 and 1, which can then be used to divide data into two classes. To fit the 75 different classes in our supervised learning problem, two approaches of fitting the logistic regression model were utilized. The first approach re-labels each class using one-vs.-all (OvA) where each class is treated as a separate binary classification problem with 1 indicating the class of interest and 0 indicating the remaining classes. This OvA strategy allows a binary logistic regression model to be fit to 75 neuronal cell types in an iterative fashion. The second approach uses the multinomial extension of logistic regression—a generalized linear model—which innately permits multiclass classification and trains the classification of the 75 classes simultaneously, producing a single model.

SVM utilizes hyper-planes to divide data into positive and negative classes by attempting to maximize the margin between the classified data on the two sides of the hyper-plane [25]. For this cell type classification problem, hyper-planes are constructed to distinguish between cell types according to high dimensional gene expression patterns. SVM also has the potential to explore nonlinear relationships in data via the use of nonlinear kernel functions. In this study, multiple kernel functions were evaluated to determine their impact on the classification efficacy.

Random forest is an ensemble machine learning method that uses the bagging technique to train a large number of decision trees to achieve higher classification accuracy than individual trees [26]. At each tree node, the data is partitioned into two groups that optimize class purity using randomly selected candidate features. For new data points, the classification predictions from the "forest" of decision trees are then combined to produce a final classification prediction by the majority vote rule [26]. In many random forest implementations, the algorithm keeps track of the features that are most informative for classification, thus retaining classification explainability [21]. Similar to logistic regression, both OvA and multiclass versions of random forest models were built.

Neural networks are deep learning methods that explore complex nonlinear structure of the data space by constructing sets of nodes (a.k.a. neurons) interconnected through a number of hidden layers to produce a directional network. Typically, neural networks are used to produce complex predictive models that are able to find nonlinear patterns in data used as the input layer [25, 26]. Each hidden layer of the neural network is defined by applying a nonlinear activation function to a linear combination of input variables from the previous input or hidden layer. Commonly used activation functions are sigmoid and ReLU. Neural networks are fitted using the backpropagation algorithm with respect to a given loss function; for classification, it is usually the cross-entropy loss function.

LightGBM is an implementation of gradient boosting machine (GBM), which is an ensemble of decision trees optimized over a differentiable loss function. Unlike random forests where the collection of decision trees are independent, GBM constructs decision trees sequentially [24]. In this way the features are sorted with Gradient-based One-Side Sampling (GOSS)

according to the training objective at each split using accumulated values to find the best split. In our application, LightGBM trained and predicted these classifications significantly faster compared to the other tested models because it utilizes histogram-based algorithms rather than pre-sort-based algorithms [24].

## Model hyperparameters

A full list of hyperparameters that were tuned for each machine learning method can be found in Table 1 with the list of optimal hyperparameters for each model found in S1 Table in S1 File. The optimal models were decided by choosing a set of hyperparameter values that resulted in the highest median F-beta value with respect to the 75 clusters using the validation dataset. Hyperparameter tuning is discussed further in the Results section.

## Statistical analysis

For performance analysis, F-beta scores were calculated from the confusion matrices that each model produced. F-beta scores are a numeric measurement of classification accuracy calculated as the weighted geometric mean of precision and recall.

$$F_\beta = (1 + \beta^2) \times \frac{precision \times recall}{\beta^2 \times precision + recall}$$

**Table 1. Machine learning hyperparameters.**

| Method | Hyperparameter | Description |
|---|---|---|
| Logistic Regression [22] | Penalty | A parameter used to specify the norm used in the penalization |
| | Solver | A parameter used to specify the method to find the parameter weights that minimize a cost function |
| | Max_iter | The maximum number of iterations taken for the solvers to converge |
| | Multi_class | A parameter that determines the fit for each label |
| | n_jobs | A parameter that represents the number of CPU cores used when parallelizing over classes |
| Support Vector Machine [22] | Kernel | Utilizes a function used to map nonlinear observations onto a higher dimensional space so they are more easily distinguished |
| | Regularization C | A penalty used to tell the algorithm how much to value the misclassification of points when training the model. The strength of the regularization is inversely proportional to C. |
| Random Forest [22] | Number of Trees | The number of decision tree classifiers built and combined to get a final random forest model |
| Neural Network [23] | Neurons per layer | The number of neurons in each layer of the network |
| | Activation Function | The activation function of a node defines the output of that node given an input or set of inputs |
| | Number of Hidden Layers | The number of layers in the network excluding the input and output layers |
| | Regularization | Type of function used that adds information in order to prevent overfitting. If p > n, the ordinary least squares estimator is not unique and will heavily overfit the data. Thus a form of complexity regularization will be necessary. |
| | Optimizer | The function that minimizes (or maximizes) the loss in order to train the model. |
| | Loss Function | Loss functions for classification are computationally feasible loss functions representing the price paid for inaccuracy of predictions in classification problems |
| LightGBM [24] | num_leaves | The maximum tree leaves for base learners to control the complexity of the model. A large num_leaves leads to better accuracy but can also result in more overfitting |
| | max_bin | The maximum number of bins to place feature values into. A large max_bin value increases accuracy while a small max_bin value helps minimize overfitting. |
| | Min_data_in_leaf | A parameter used to prevent over-fitting in a leaf-wise tree. Its optimal value depends on the number of training samples and num_leaves |
| | lambda_l2 | This stands for L2 regularization and is used to minimize overfitting |
| | extra_trees | Uses randomized trees |
| | path_smooth | Controls smoothing applied to the nodes of trees. |

Here, the beta ($\beta$) parameter of the equation controls the tradeoff between precision and recall. For our study we used $\beta = 0.5$ in order to provide less weight to the recall value to control for the technical dropout artifact observed in sc/snRNA-seq experiments as in a previous study [21]. Higher F-beta scores indicate that more nuclei were correctly classified.

When comparing the validation F-beta scores between different CV thresholds (S2 Table in S1 File) and between the default and optimal models (S3 Table in S1 File), Wilcoxon signed-rank test was used to determine if there are statistically significant differences between the paired sets of F-beta scores.

### Kidney dataset

Model performances were independently evaluated on a second dataset using the same methodologies as the human MTG snRNA-seq models. The kidney snRNA-seq dataset from the Human BioMolecular Atlas Program (HuBMAP) Consortium consists of 29,732 gene-level counts and 64,693 samples with each sample classified under one of 49 cell types [27]. The same data pre-processing steps were performed on the kidney dataset to normalize the counts and then split into a 60%/20%/20% training/validation/testing split. Four machine learning models were built on the kidney dataset–binary and multinomial Logistic Regression models, Neural Networks, and LightGBM–with each model utilizing the optimal hyperparameters found for the MTG data as described in S1 Table in S1 File.

### Code availability

The code used to preprocess the data and run each model can be found on this GitHub repository: https://github.com/HuyGLe/neuronal-cell-type-classification.

## Results

### Feature selection

Different gene subsets derived from different feature selection approaches were used as inputs for machine learning model construction. Two different metrics for feature selection were explored for the MTG dataset–coefficient of variation (CV), as a measure of general expression variability, and binary expression score (BIN), as a measure of cluster specificity. For MTG CV, thresholds of 0.52, 1.5, 2.5, 3.5, and 4.5 produced 11,189, 6,063, 3,048, 1,426, and 709 genes, respectively. For MTG BIN, using the top 15%, 10%, 5%, and 1% genes ranked by BIN produced 13,445, 12,604, 12,029, and 4,741 genes, respectively. Note that even with the small percentages of the top ranking BIN genes within each cluster, the final combined list of BIN genes remains relatively large compared to the high CV thresholds. This indicates that there is little overlap between the top BIN ranked genes identified for each cluster, which makes sense given that the BIN method selects for genes that are cluster specific.

For binary logistic regression, SVM, random forests, and neural networks, BIN gene sets consistently produced lower F-beta scores than CV gene sets, while LightGBM and multinomial logistic regression displayed similar results for both feature selection methods (Fig 2). LightGBM and multinomial logistic regression were also the most robust to different feature selection thresholds. In contrast, binary logistic regression, SVM, and neural networks showed better average F-beta values using higher CV and BIN thresholds, suggesting these methods were not able to select the most useful features when more features were retained at lower threshold values. Multiclass random forests + CV model also showed similar trends, however the random forests + BIN F-beta values decreased with more stringent thresholds. This suggests that the random forests' own feature selection criteria may be superior to the feature

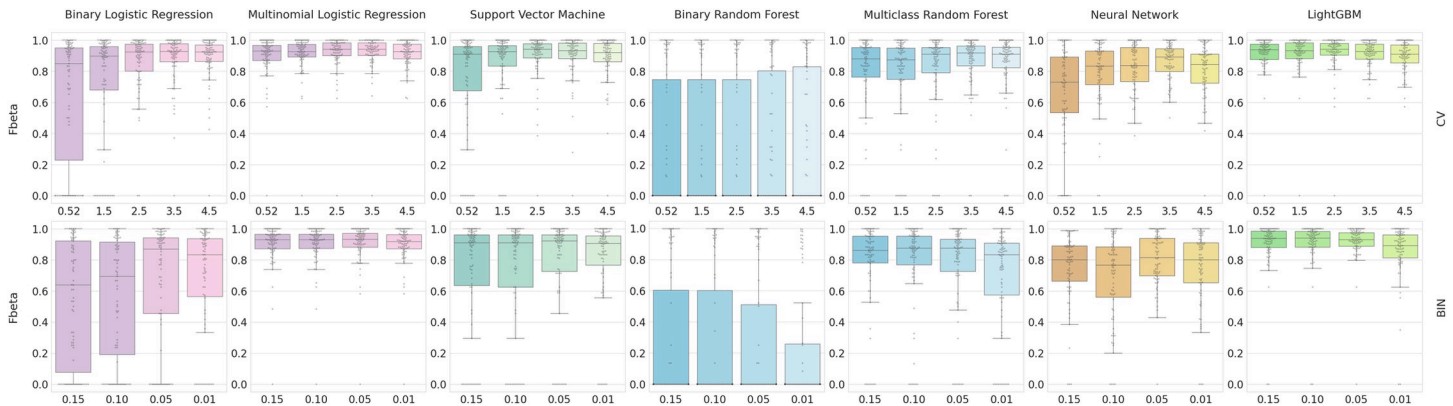

**Fig 2. Classification performance using two feature selection methods (CV vs BIN filtering) and five machine learning methods.** For feature selection using coefficient of variation (CV), the filtering thresholds from left to right were 0.52, 1.5, 2.5, 3.5, and 4.5. For binary score (BIN), the filtering thresholds from left to right are 0.15, 0.10, 0.05, and 0.01. The resulting number of genes for each threshold is listed below the threshold labels. Each of these feature sets was used by five different machine learning methods (LightGBM, Neural Network, SVM, Logistic Regression, Random Forest) using the training data. F-beta was calculated as a measure of classification accuracy.

selection methods evaluated here. CV threshold of 3.5 was optimal for logistic regression, random forests, and neural networks, while 2.5 was optimal for LightGBM and SVM (Fig 2). High CV thresholds also showed fewer outliers with F-beta scores of 0, which came predominantly from smaller cluster sizes where there may not have been enough informative marker genes in the training data to adequately train for these small clusters. Varying values of BIN thresholds showed no significant pattern regardless of the method used. The BIN threshold of 1% was optimal for SVM and logistic regression, 5% for neural networks, and 10% for LightGBM and random forests (Fig 2). A comparison of the classification performance using different CV thresholds in shown in S2 Table in S1 File.

## Hyperparameter optimization

Hyperparameters for each method were adjusted to produce the highest average F-beta scores on the validation dataset. The Wilcoxon signed-rank test was used to determine if there was a statistically significant difference between the default and the optimal hyperparameter settings that produced the best F-beta values for each cluster with the least outliers when tested in validation (S3 Table in S1 File). LightGBM showed the biggest difference (lowest p-value) between default and optimal hyperparameter settings, followed by neural networks, SVM, random forests and logistic regression. For most of the models, optimal hyperparameters (shown in color) resulted in superior performance compared to default hyperparameters (shown in gray) with logistic regression as the exception (Fig 3). In all optimal models, cluster sizes between 2 [4] (16) and 2 [8] (256) showed more variability in performance suggesting all methods performed worse with smaller cluster sizes (with the exception of the smallest cluster corresponding to endothelial cells). The full list of optimal hyperparameters for each machine learning method can be found in S1 Table in S1 File. The full list of unique hyperparameter combinations tested along with their corresponding F-beta results can be found on the "results" folder of the GitHub repository: https://github.com/HuyGLe/neuronal-cell-type-classification/tree/main/results.

For all methods, models including regularization performed better than those without, with the default values (C = 1) generally showing the best results, except for neural networks, which required a relatively small value (C = 1e-6) (data not shown). Specifically for logistic regression and LightGBM, L2 regularization models produced better results than L1 regularization models. Although rbf is a popular kernel choice for SVM models, linear kernels (i.e., no kernel)

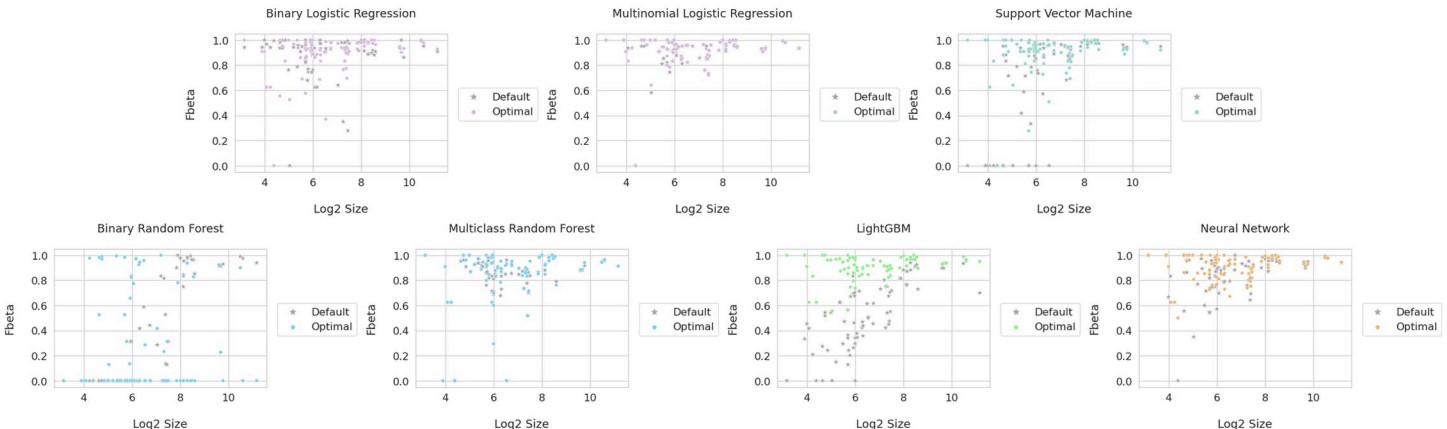

**Fig 3. Classification performance between the default and optimal hyperparameter settings.** The 3.5 CV feature set was used and models produced using default and optimal hyperparameter setting. F-beta was calculated as a measure of classification accuracy. Log2 size is log base 2 of the cluster size. The labeled p-values for each method are from the Wilcoxon signed-rank test between the default and optimal validation.

produced higher F-beta values, especially for smaller cluster sizes. In neural networks, our study found that as long as the number of neurons per layer did not exceed the number of features, the number of neurons had little impact on performance, and therefore 100 was chosen as a consistent value. Neural network models with fewer than 3 hidden layers produced poor results; networks with a larger number of layers simply increased runtime with little to no difference in results. All scikit-learn combinations of activation functions and solvers were tested with ReLU and Adam producing the best results for this dataset. For LightGBM, minimizing overfitting by hyperparameter adjustment was critical, as LightGBM models using default settings provided perfect F-beta scores when predicted on the training set but poor performance on the validation set. Thus, following overfitting recommendations from LightGBM documentation, min_data_in_leaf and lambda_l2 values were increased significantly from default values with both extra_trees and path_smooth parameters turned on.

Both one-vs.-rest and multiclass versions of logistic regression and random forest models were compared, with multiclass versions providing superior results compared to the binary versions and requiring significantly less time to train.

## Overfitting

Overfitting is a major problem when training supervised machine learning algorithms producing models that do not perform well when given new data. Overfitting may occur when noisy features correlate with class membership by chance in the training datasets or when model complexity exceeds the data complexity. Indeed, many of the tuned hyperparameters, for example regularization, were optimized mainly to reduce the impact of overfitting on the validation dataset as noted above.

To assess the impact of overfitting, optimal models for each method were utilized to predict on the training, validation, and testing datasets (Fig 4) with the differences in F-beta score distributions for the three datasets showing the effects of overfitting. Although some differences between training and testing results is generally expected when the number of features exceeds the number of samples, it is clear that the training results are far superior to both the validation and test results, as optimal models produced nearly perfect classification accuracy using the training data for all methods but lower F-beta score on the validation and test datasets. To assess the impact of overfitting, the F-beta classification accuracy scores between the training,

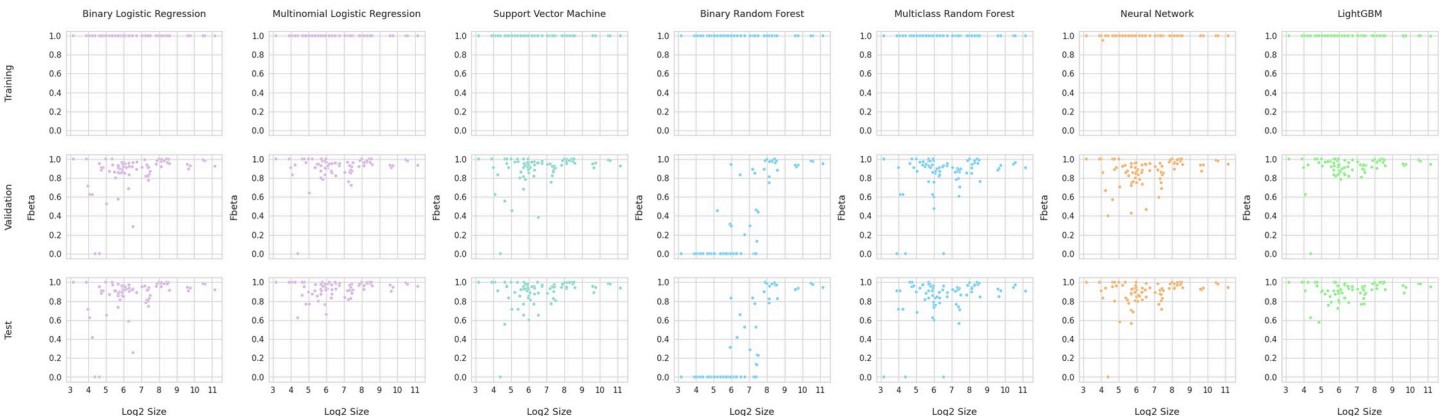

**Fig 4. Model performance on training, validation, and test datasets.** CV thresholds of 2.5 were used for Multinomial Logistic Regression and Neural Networks while a threshold of 3.5 was used for all other models. F-beta was calculated as a measure of classification accuracy for the training, validation, and test datasets. Log2 size is log base 2 of the cluster size. Differences between these distributions highlight the effect of overfitting.

validation, and test datasets were compared using the Wilcoxon signed-rank test (See S3 Table in S1 File). Large differences between training and validation performance are indicated by the low p-values. No significant differences were observed between the validation and test datasets for any of the optimally-configured machine learning methods.

Overall, multinomial logistic regression and LightGBM consistently outperformed the other machine learning methods with SVM, random forests, and neural networks performing similarly to each other. Logistic regression, however, produced slightly superior results on the test dataset compared to LightGBM (Fig 5). All optimized models, excluding binary random forest, had median F-beta scores above 0.9 for this test dataset. Classification using multinomial logistic regression produced F-beta scores that were all above 0.6.

## Model performance on the kidney dataset

For the kidney dataset, the coefficient of variation (CV) approach was used with thresholds of 1.0, 1.5, 2.5, 3.5, 4.5, 5.5, 6.6, and 7.5 to select informative features. Given the larger size of the

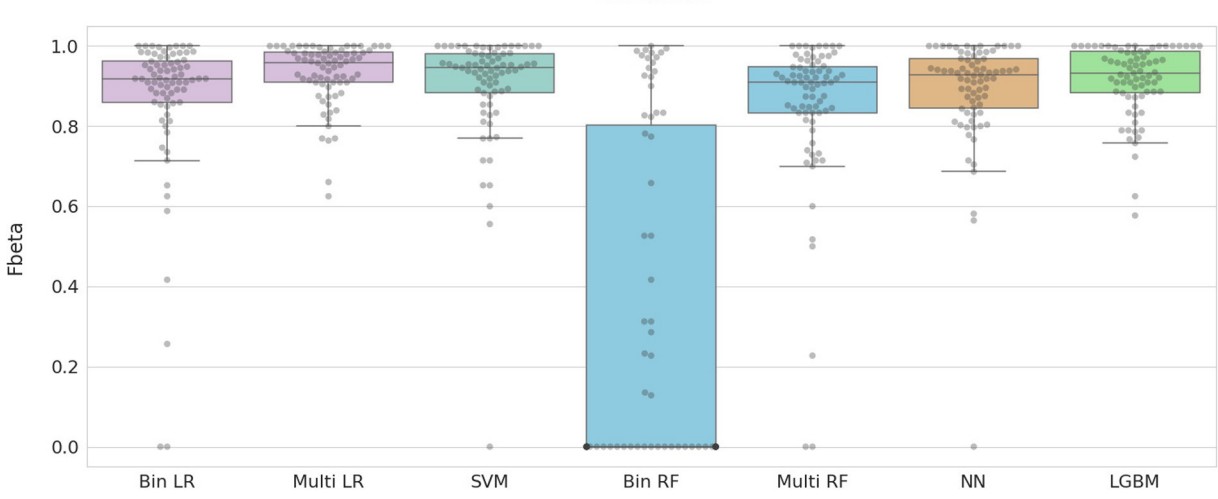

**Fig 5. Classification performance for seven supervised machine learning models with optimal feature selection and hyperparameters.** F-beta was calculated as a measure of classification accuracy. See S1 Table in S1 File for optimal hyperparameter settings.

kidney dataset compared to the MTG dataset, the larger CV thresholds (5.5 and above) were added in order to produce similar numbers of genes as input features as with the MTG analysis. The binary expression score (BIN) approach was not applied because the CV approach led to better classification results with the MTG dataset as discussed above. The CV thresholds resulted in 16,637, 14,438, 11,070, 8,594, 6,508, 4,572, 2,386, and 588 genes, respectively.

Based on the model performances with the MTG dataset, four supervised machine learning models (two top performers–multinomial logistic regression and LightGBM, and binary logistic regression as a sibling model in the logistic regression family and the deep learning neural network method) were selected for evaluation using the kidney dataset. Fig 6 shows model performances on the kidney dataset using the optimal hyperparameters trained from the MTG dataset. Similar to the MTG dataset, the top performers for classifying kidney cell types were the multinomial logistic regression and LightGBM, both with the optimal CV threshold at 1.5. The best median F-beta values using the kidney dataset were slightly lower than those obtained using the MTG dataset across all models. This could be explained by the fact that the kidney dataset was modelled using the hyperparameters tuned to the MTG dataset.

The models built using the MTG dataset showed a trend toward higher F-beta values at higher CV thresholds. However, models built using the kidney dataset did not show the same trend. All models built on the kidney dataset exhibited a downtrend in the F-beta values at higher CV thresholds, with significant dropoff beyond the 5.5 threshold. The classification models for the kidney dataset achieved the best median F-beta values at low CV thresholds (1.0 for binary logistic regression and 1.5 for the other three models), suggesting it requires more gene features to distinguish kidney cell types compared to the neuronal cell types. However, the relative accuracy of the different models with only slight differences between the best median F-beta values of these two datasets, suggesting that the cell type classification models using supervised machine learning methods are robust and not dependent on a single dataset.

## Discussion

The results presented here suggest that several different machine learning (ML) methods and input feature sets are effective at producing models for the classification of cell types from single nucleus RNA sequencing-based transcriptional profiling, with all five ML methods producing median F-beta score above 0.9 for the 75 cell type clusters under optimal configuration conditions in the set aside test dataset. While both coefficient of variation and binary

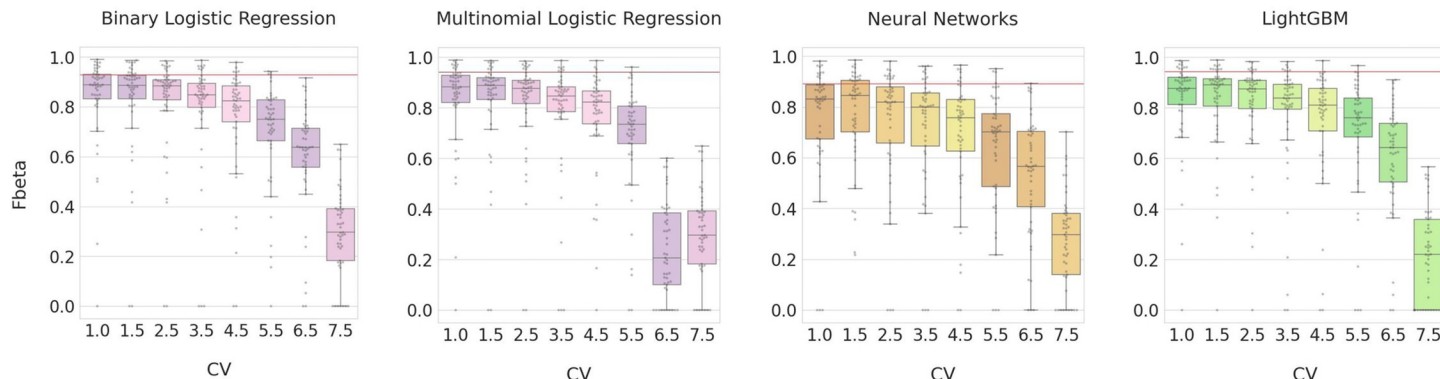

**Fig 6. Model performance on the kidney dataset.** Four models (Binary Logistic Regression, Multinomial Logistic Regression, Neural Networks, and LightGBM) were built using the optimal hyperparameters trained from the MTG dataset. Eight CV thresholds (1.0, 1.5, 2.5, 3.5, 4.5, 5.5, 6.5, and 7.5) were applied for feature selection. The red horizontal line indicates the median F-beta value for the best CV threshold for a given method from the MTG dataset.

expression score produced sets of useful input gene expression features for model construction at optimal thresholds for all five ML methods, some methods were more sensitive to the number of features available when different thresholds were used (e.g., SVM and neural networks) suggesting that the feature selection approach inherent in these ML methods were less effective in identifying the most useful feature in the larger input feature sets. Likewise, some methods were much more sensitive to hyperparameter setting than others. For example, the LightGBM approach run using default settings produced poor classification models, which were dramatically improved with hyperparameter optimization using the validation dataset.

Model overfitting had a significant impact on model performance. All five ML methods produced virtually perfect classification results (F-beta = 1) on the datasets used for training but invariably showed reduced performance on both the validation set used for hyperparameter optimization and the set-aside dataset used for testing, suggesting a problem with overfitting. This is probably not too surprising given the large number of input features available even with the more stringent threshold settings. This suggests that the expression of many of these features are noisy and just happen to correlate with cell cluster membership by chance in the training dataset. Indeed, feature selection is an extremely fundamental and yet often overlooked step in the machine learning processing pipeline. In this case, the coefficient of variation at the 2.5 and 3.5 cutoffs gave optimal results presumably by removing uninformative features (i.e., variable housekeeping genes) that were adding noise to the models. Selecting optimal hyperparameters to avoid overfitting improved model performance to some degree, but there appears to be a limit as to how much these parameters can control overfitting. Interestingly, model performance on both the validation dataset and the testing dataset was similar, suggesting that the informative classification genes were informative for both and the noisy correlated genes were noisy for both. Thus, approaches that could distinguish between true biological correlations and chance correlations during feature selection could have a positive impact on model performance on independent datasets and would be worth exploring further. As feature selection criteria may be different for every dataset, applying different feature selection approaches and assessing their performance would be important for classification model construction.

As a benchmark for comparison, we compared the F-beta classification scores of the top performing ML method, multinomial logistic regression, with those obtained using the minimum set of marker genes produced by NS-Forest [21] and reported in S3 Table in S1 File of the primary publication [5]. Multinomial linear regression models consistently outperformed the minimum NS-Forest markers genes (See S3 Fig in S1 File). NS-Forest was designed to find the minimal set of markers to define each cell type, resulting in 157 marker genes for the 75 cell type clusters. This is significantly less than the gene sets with the CV and BIN feature selection, even at the most stringent thresholds (the minimum sets containing 709 and 4,741 genes, respectively). Thus, there is a clear tradeoff between minimizing the number of genes and accurate classification.

Finally, one of the more surprising observations was the relatively poor performance of random forest machine learning when run in binary classification mode. Given that the random forest approach is one of the more popular machine learning methods and yet showed such a dramatic difference in classification performance when run in the binary classification mode in comparison with all other methods or when run in multiclass mode, finding a suitable hypothesis for the observed phenomenon was warranted. One possible explanation relates to how important features are selected during the decision tree process and how the results from the "forest" of trees are compiled. In binary classification, different random forest models are produced for each cell type cluster by comparing the gene expression patterns in nuclei for Cluster X with the gene expression patterns in all nuclei from all the non-X clusters combined.

To build each decision tree, a random set of gene expression features is first selected (in this case, the number of gene expression features selected was the square-root of the total number of gene expression features available), which are then used to identify the features best able to separate nuclei from Cluster X from nuclei from all other clusters. In the case of the use of CV to select input features, there is no guarantee that any given gene expression feature will be good for distinguishing Cluster X from non-Cluster-X. Indeed, the vast majority of features would probably not be ideal from classifying Cluster X. Even with the BIN approach, since binary scoring selection is spread across all 75 clusters, the chances that a good marker for Cluster X is selected at random in any given decision tree is probably small. Therefore, most decision trees will be constructed from randomly selected gene expression features that are sub-optimal for classification of Cluster X. And since these relatively poor decision trees are treated equally with relatively good trees for the final random forest classification result, they would have a negative impact on classification performance.

In contrast, when run in multiclass mode, as long as an available gene expression feature is good for classifying any one of the 75 cell type clusters, it will be selected for the first branch point in the decision tree to segregate that cluster. And if the next available gene expression feature is good for classifying any other cell type cluster, it will be chosen next and contribute positively to the multiclass decision tree. Thus, the chance that a useful feature is available in the randomly selected gene expression feature list increases approximately 75-fold and the overall quality of the forest of trees would be expected to be much higher.

Using the kidney dataset as an independent testing dataset, LightGBM and multinomial logistic regression were the models that showed superior performance compared to binary logistic regression and neural networks, as was observed with the MTG dataset. Of these machine learning methods for classifying cell types using snRNA-seq data, the models evaluated on the kidney data showed robust performance given an effective feature selection method. Although the trend of higher CV thresholds performing better was not observed in the kidney dataset, low CV thresholds (for example 1.0, 1.5 and 2.5) still produced excellent F-beta performance. It's reasonable to assume that there may be other more effective feature selection approaches that could be developed and assessed for this purpose.

## Conclusion

In conclusion, this study has shown that machine learning is capable of producing models for classifying cell types based on gene expression profiles produced by single cell/nucleus transcriptomic data. The results presented demonstrate the effects of input feature selection, hyperparameter tuning, and machine learning method type on cell type classification performance. The resulting optimized workflows are not limited to neuronal cell types but have the potential to produce classification models for any cell type in any tissue from any species. The reduced performance observed on the validation and test datasets suggest that while hyperparameter tuning improved model performance overall, overfitting with noisy features remains a problem. Thus, exploring alternative methods for distinguishing between truly useful classification features and noisy correlated features could further improve the performance of machine learning models constructed for cell type classification. The increasing application of high-throughput single cell sequencing technologies and advanced machine learning analysis promises to revolutionize our understanding of how cellular heterogeneity contributes to overall tissue function.

## Supporting information

**S1 File.**
(PDF)

## Acknowledgments

This work was performed as part of the Bioengineering Senior Design Project for H.L., B.P., J. U., and D.C. The authors would like to thank the Allen Institute for Brain Science for providing the data for this study. We thank Dr. Bruce Wheeler and Behrad Tagdiri for their guidance throughout the whole Senior Design Project. We thank Yu (Max) Qian and Padhraic Smyth for their critical review of the manuscript prior to submission.

## Author Contributions

**Conceptualization:** Yun Zhang, Brian D. Aevermann, Richard H. Scheuermann.

**Data curation:** Richard H. Scheuermann.

**Formal analysis:** Yun Zhang, Brian D. Aevermann.

**Investigation:** Huy Le, Beverly Peng, Janelle Uy, Daniel Carrillo.

**Methodology:** Huy Le, Beverly Peng, Janelle Uy, Daniel Carrillo, Yun Zhang, Brian D. Aevermann.

**Project administration:** Richard H. Scheuermann.

**Software:** Huy Le, Beverly Peng, Janelle Uy, Daniel Carrillo.

**Supervision:** Richard H. Scheuermann.

**Validation:** Janelle Uy.

**Visualization:** Huy Le, Beverly Peng.

**Writing – original draft:** Huy Le, Beverly Peng, Janelle Uy, Daniel Carrillo.

**Writing – review & editing:** Yun Zhang, Brian D. Aevermann, Richard H. Scheuermann.

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
