## [Decision Letter · Decision Letter 0]

24 Mar 2022

PONE-D-22-03357Machine Learning for Neuronal Cell Type Classification from Single Nucleus RNA Sequencing DataPLOS ONE

Dear Dr. Scheuermann,

Thank you for submitting your manuscript to PLOS ONE. After careful consideration, we feel that it has merit but does not fully meet PLOS ONE’s publication criteria as it currently stands. Therefore, we invite you to submit a revised version of the manuscript that addresses the points raised during the review process. Both reviewers expressed some concerns on specific aspects of the manuscript and suggested some further analyses that could help strengthening your argument. Please consider all these suggestions and either perform the required analysis or discuss why it is not necessary or feasible. The reviews also contain suggestions for further references to be cited that you should consider.  

We look forward to receiving your revised manuscript.

Kind regards,

Paolo Provero, Ph.D.

Academic Editor

PLOS ONE

Journal Requirements:

2.Please include captions for your Supporting Information files at the end of your manuscript, and update any in-text citations to match accordingly. Please see our Supporting Information guidelines for more information: http://journals.plos.org/plosone/s/supporting-information. 

Reviewers' comments:

Reviewer's Responses to Questions

**Comments to the Author**

1. Is the manuscript technically sound, and do the data support the conclusions?

Reviewer #1: Partly

Reviewer #2: Partly

2. Has the statistical analysis been performed appropriately and rigorously? 

Reviewer #1: Yes

Reviewer #2: Yes

3. Have the authors made all data underlying the findings in their manuscript fully available?

Reviewer #1: Yes

Reviewer #2: Yes

4. Is the manuscript presented in an intelligible fashion and written in standard English?

Reviewer #1: Yes

Reviewer #2: Yes

5. Review Comments to the Author

Reviewer #1: In this paper the authors worked on the evaluation of machine learning models in the task of single cell data cell type classification. The tested dataset is characterised by the expression data for a relatively huge sample, previously annotated in 75 different cell types. The authors evaluated two approaches for the features selection (binary expression score and coefficient of variation) and several hyperparameter for each model. Classification accuracy and models comparison were done by evaluating the F-beta score.

Overall, they showed how the combination of feature selection and the right model (with the correct hyperparameters) can be used for cell type classification.

Here some points the authors should address:

1) The main purpose of the paper as they stated in the Introduction is the comparison of models and feature selection on the classification task. The type of biological data they analysed (MTG of human brain) is irrelevant to this end. To have more robust conclusions from the modeling part they should apply the same procedure to another dataset.

2) Could be of interest to see how the 75 cell types are divided: are there cell types more represented than others? And if it is true, is the overrepresentation of a cell type correlated with the model performance?

3) They should describe in more details the hyperparameters tuning section, as it is a main point of the paper. In table 1 (that could also be moved in the supplementary) they just described what hyperparameters they would look at. There is no paragraph in the paper where they describe how they changed the parameter (for example, in the neural network model, how many hidden layers were tested? In the random forest, which number of trees were used?). In the end, how many models were trained?

4) The overfitting showed from the models is worrying. This is also a reason to test the approaches on another dataset. I would suggest to evaluate the model on a cross-validation setting and if this does not change the outcome to try to justify the overfitting with the composition of the data (number of cells per cell type).

5) The github page is missing the notebook for the neural network implementation. I would also suggest to improve the readme with a bit of introduction and explanation of the folders.

Reviewer #2: Cell phenotype classification is an important task. Single-cell genomic data enable researchers to conduct this analysis based on similarity of cell's gene expression profile. The authors consider this as a standard supervised machine learning problem, and compared a few general-purpose machine learning methods' performance in cell-phenotype-classification. One data set is used (split into training, validation, and test sets) in this benchmark study.

Following are my major concerns:

1. The literature is not well discussed. For example:

1.1 A similar benchmark study was recently published: Cao, X. et al. A Systematic Evaluation of Supervised Machine Learning Algorithms for Cell Phenotype Classification Using Single-Cell RNA Sequencing Data. Frontiers Genetics 13, 836798 (2022).

Authors should properly cite such related work and discuss the difference or the added contribution by their work.

1.2 Besides general-purpose machine learning methods, there are other supervised learning methods tailored particularly for cell phenotype classification. For example, the following paper utilizes the proportion of dropout as a useful feature to help classification

Ji et al. scAnnotate: an automated cell type annotation tool for single-cell RNA-sequencing data. bioRxiv 1–9 (2022) doi:10.1101/2022.02.19.481159.

I suggest including a few methods like this in the evaluation, or at least properly discuss existence of such related work.

2. In the evaluation, authors selected F-beta score for beta=0.5, which is a quite subjective choice. I suggest authors evaluate performance using other beta values to see how beta value affect the results of evaluation.

3. The figure quality is bad. I can barely see what's in Figure 2,3,4. Authors need to provide higher resolution files and make the text in figure large enough to read.

6. PLOS authors have the option to publish the peer review history of their article (what does this mean?). If published, this will include your full peer review and any attached files.

Reviewer #1: No

Reviewer #2: No

---

## [Decision Letter · Decision Letter 1]

16 Aug 2022

PONE-D-22-03357R1Machine Learning for Cell Type Classification from Single Nucleus RNA Sequencing DataPLOS ONE

Dear Dr. Scheuermann,

Thank you for submitting your manuscript to PLOS ONE. After careful consideration, we feel that it has merit but does not fully meet PLOS ONE’s publication criteria as it currently stands. Therefore, we invite you to submit a revised version of the manuscript that addresses the points raised during the review process. As you can see, reviewer 2 objects that you use a single dataset, but this problem has been fixed in your revision, therefore you can safely ignore this objection. The same reviewer also suggests some additional papers to be included in the references, that you might want to consider. I do not consider as mandatory the inclusion of these references.  

We look forward to receiving your revised manuscript.

Kind regards,

Paolo Provero, Ph.D.

Academic Editor

PLOS ONE

Journal Requirements:

Reviewers' comments:

Reviewer's Responses to Questions

**Comments to the Author**

1. If the authors have adequately addressed your comments raised in a previous round of review and you feel that this manuscript is now acceptable for publication, you may indicate that here to bypass the “Comments to the Author” section, enter your conflict of interest statement in the “Confidential to Editor” section, and submit your "Accept" recommendation.

Reviewer #1: All comments have been addressed

Reviewer #2: (No Response)

2. Is the manuscript technically sound, and do the data support the conclusions?

Reviewer #1: Yes

Reviewer #2: (No Response)

3. Has the statistical analysis been performed appropriately and rigorously? 

Reviewer #1: Yes

Reviewer #2: No

4. Have the authors made all data underlying the findings in their manuscript fully available?

Reviewer #1: Yes

Reviewer #2: Yes

5. Is the manuscript presented in an intelligible fashion and written in standard English?

Reviewer #1: Yes

Reviewer #2: Yes

6. Review Comments to the Author

Reviewer #1: (No Response)

Reviewer #2: Authors use one data set to compare multiple machine learning methods and found multinomial logistic regression work best.

There are two critical issues. I list them below

First, literature review is poorly done. Authors should know the field before they start this work. Those works should be properly cited and discuss authors new contributions added to existing work.

They should start with a review paper

1.Pasquini, G., Arias, J., Schäfer, P. & Busskamp, V. Automated methods for cell type annotation on scRNA-seq data. Comput Struct Biotechnology J 19, 961–969 (2021).

There are very comprehensive benchmark papers for specific cell type classfication, such as

2.Abdelaal, T. et al. A comparison of automatic cell identification methods for single-cell RNA sequencing data. Genome Biol 20, 194 (2019).

There are also benchmark paper for comparing machine learning methods on cell annotation using scRNA-seq data

3.Cao, X. et al. A Systematic Evaluation of Supervised Machine Learning Algorithms for Cell Phenotype Classification Using Single-Cell RNA Sequencing Data. Frontiers Genetics 13, 836798 (2022).

Second, using one data set to claim one method is better is not comprehensive. Authors should change their statement to make it weak enough or use more data sets to check robustness of their comparison.

7. PLOS authors have the option to publish the peer review history of their article (what does this mean?). If published, this will include your full peer review and any attached files.

Reviewer #1: No

Reviewer #2: No

---

## [Author Response · Author response to Decision Letter 1]

1 Sep 2022

Response to Reviewers

The relevant changes made to the manuscript in response to the reviewers’ comments are underlined in the “Tracked Changes” version. A summary of the response to each reviewer concern is summarized in bold italic below.

Reviewer # 1

No changes requested.

Reviewer #2

Authors use one data set to compare multiple machine learning methods and found multinomial logistic regression work best.

There are two critical issues. I list them below

First, literature review is poorly done. Authors should know the field before they start this work. Those works should be properly cited and discuss authors new contributions added to existing work.

They should start with a review paper

1.Pasquini, G., Arias, J., Schäfer, P. & Busskamp, V. Automated methods for cell type annotation on scRNA-seq data. Comput Struct Biotechnology J 19, 961–969 (2021).

There are very comprehensive benchmark papers for specific cell type classfication, such as

2.Abdelaal, T. et al. A comparison of automatic cell identification methods for single-cell RNA sequencing data. Genome Biol 20, 194 (2019).

There are also benchmark paper for comparing machine learning methods on cell annotation using scRNA-seq data

3.Cao, X. et al. A Systematic Evaluation of Supervised Machine Learning Algorithms for Cell Phenotype Classification Using Single-Cell RNA Sequencing Data. Frontiers Genetics 13, 836798 (2022).

We have expanded the Introduction section (page 3 – 4) to include a summary of these relevant publications and how our study differs. Also included in Literature citation

Second, using one data set to claim one method is better is not comprehensive. Authors should change their statement to make it weak enough or use more data sets to check robustness of their comparison.1. 

We added the analysis of a second dataset derived from human kidney with a similar level of cellular complexity showing similar performance characteristics. This had been added as the new Figure 6 and described in the Methods (page 8), Results (page 11 – 12), and Discussion (page 14) sections.

---

## [Editor Report · Decision Letter 2]

12 Sep 2022

Machine Learning for Cell Type Classification from Single Nucleus RNA Sequencing Data

PONE-D-22-03357R2

Dear Dr. Scheuermann,

We’re pleased to inform you that your manuscript has been judged scientifically suitable for publication and will be formally accepted for publication once it meets all outstanding technical requirements.

Kind regards,

Paolo Provero, Ph.D.

Academic Editor

PLOS ONE
---

## [Editor Report · Acceptance letter]

13 Sep 2022

PONE-D-22-03357R2 

Machine Learning for Cell Type Classification from Single Nucleus RNA Sequencing Data 

Dear Dr. Scheuermann:

I'm pleased to inform you that your manuscript has been deemed suitable for publication in PLOS ONE. Congratulations! Your manuscript is now with our production department. 

Kind regards, 

on behalf of

Dr. Paolo Provero 

Academic Editor

PLOS ONE